# Host-induced spermidine production in motile *Pseudomonas aeruginosa* triggers phagocytic uptake

Sebastian Felgner[1†], Matthias Preusse[1†], Ulrike Beutling[2], Stephanie Stahnke[3], Vinay Pawar[1], Manfred Rohde[4], Mark Brönstrup[2], Theresia Stradal[3], Susanne Häussler[1,5,6,7]*

[1]Department of Molecular Bacteriology, Helmholtz Centre for Infection Research, Braunschweig, Germany; [2]Department of Chemical Biology, Helmholtz Centre for Infection Research, Braunschweig, Germany; [3]Department of Cell Biology, Helmholtz Centre for Infection Research, Braunschweig, Germany; [4]Central Facility for Microscopy, Helmholtz Centre for Infection Research, Braunschweig, Germany; [5]Department of Molecular Bacteriology, Twincore, Hannover, Germany; [6]Department of Clinical Microbiology, Rigshospitalet, Copenhagen, Denmark; [7]Cluster of Excellence RESIST (EXC 2155), Hannover Medical School, Hannover, Germany

*For correspondence: susanne.haeussler@helmholtz-hzi.de

[†]These authors contributed equally to this work

Competing interests: The authors declare that no competing interests exist.

**Abstract** Exploring the complexity of host–pathogen communication is vital to understand why microbes persist within a host, while others are cleared. Here, we employed a dual-sequencing approach to unravel conversational turn-taking of dynamic host–pathogen communications. We demonstrate that upon hitting a host cell, motile *Pseudomonas aeruginosa* induce a specific gene expression program. This results in the expression of spermidine on the surface, which specifically activates the PIP$_3$-pathway to induce phagocytic uptake into primary or immortalized murine cells. Non-motile bacteria are more immunogenic due to a lower expression of *arnT* upon host-cell contact, but do not produce spermidine and are phagocytosed less. We demonstrate that not only the presence of pathogen inherent molecular patterns induces immune responses, but that bacterial motility is linked to a host-cell-induced expression of additional immune modulators. Our results emphasize on the value of integrating microbiological and immunological findings to unravel complex and dynamic host–pathogen interactions.

## Introduction

A comprehensive understanding of host–pathogen interactions is crucial to advance treatment and prevention of infectious diseases. The problematic opportunistic pathogen *Pseudomonas aeruginosa* is a major cause of infection in patients with underlying or immunocompromising conditions. The opportunistic pathogen may infect virtually any tissue and has evolved as a model to study bacterial adaptation to the conditions within the human host. Therapeutic options of *P. aeruginosa* infections are increasingly limited due to the continued rise in antimicrobial resistances (*Moradali et al., 2017*; *Streeter et al., 2016*). Furthermore, the ability of the pathogen to build biofilms and to persist e.g. in the lungs of cystic fibrosis (CF) patients facilitates the establishment of chronic infections, which are largely recalcitrant to antimicrobial therapies. The development of novel drugs, vaccines and other therapeutics that work even in increasingly prevalent multidrug resistant bacteria will be highly dependent on new knowledge gained from investigating host–pathogen interactions (*Smith et al., 2017*; *Wagner et al., 2016*). The clinical course and severity of an infectious disease depends on the

results of the battle between the host's immune responses, and the pathogen's pathogenicity and virulence traits. Bacteria express pathogen-associated molecular patterns (PAMPs) such as LPS or flagella (*Janssens and Beyaert, 2003*; *Molloy, 2013*; *Mukherjee et al., 2016*; *Park et al., 2009*). Their recognition by specific host receptors called pattern recognition receptors (PRR) induces a strong inflammatory response to kill the pathogen. However, clearance of an infection is not only dependent on the ability of the host to induce an immune response following pathogen recognition, but also on the strategies of the pathogen to evade immune defense mechanisms and to express pathogenicity factors allowing them to establish a niche in the host.

Clearly, for a comprehensive understanding of host–pathogen interactions, knowledge of the behaviors of both the host and the pathogen, and their specific interaction is vital. It has become increasingly evident that host–pathogen interactions are not restricted to a linear sequence of action and reaction, but must involve a more complex crosstalk comprising entire reaction networks to determine the outcome of the disease. A prominent example of such complex crosstalk is linked to the expression of the bacterial flagellum. Flagella are essentially required for the bacterium to reach target sites in the host and to establish an infection (*Duan et al., 2013*). However, the flagellum is also recognized by toll-like receptor 5 (TLR5) or intracellular NLRC4 as a PAMP, thereby limiting bacterial establishment of an infection (*Haasken and Sutterwala, 2013*; *Yoon et al., 2012*). More recently, it has been argued that bacterial motility may play a vital role in mediating the initiation of specific immune responses and the release of inflammatory cytokines. *P. aeruginosa* strains lacking a functional flagellum have been demonstrated to undergo less phagocytosis and are more resistant to clearance in vitro and in vivo (*Amiel et al., 2010*; *Lovewell et al., 2014*; *Lovewell et al., 2011*). The lower phagocytosis rate of these non-motile strains was explained by diminished activation of the phosphatidylinositol-(3,4,5)-triphosphate (PIP$_3$) pathway in host cells (*Demirdjian et al., 2018*).

Here, we set out to further decipher the roles of flagella-dependent motility in the opportunistic pathogen *P. aeruginosa* on its interaction with the eukaryotic host. By integrating in vitro macrophage phagocytosis assays with a dual-sequencing approach (*Westermann et al., 2012*), we demonstrate that motile cells induce the expression of bacterial polyamine surface structures upon host-cell contact, while non-motile cells do not. Bacterial polyamines activate the PIP$_3$ pathway to trigger phagocytic uptake. Our results highlight novel, unforeseen metabolic components that control the molecular crosstalk between motile *P. aeruginosa* isolates and the eukaryotic host resulting from complex and dynamic actions and reactions on both the pathogen and the host side.

## Results

### Selection of flagella/motility variants of *P. aeruginosa*

We studied host–cell interactions of the mono-flagellated, fully motile *P. aeruginosa* reference strain PA14 in comparison to three isogenic motility mutants. All strains were provided by Brent Berwin (*Amiel et al., 2010*). In addition to the PA14 wild-type we obtained: (i) a Δ*fliC* mutant, which is non-motile and does not produce a flagellum due to the lack of FliC molecules that build up the filament as the major extracellular part of the flagellum; (ii) a Δ*flgK* mutant, which is non-motile and does not produce a flagellum but secretes monomeric FliC molecules that function as PAMPs (*Gillen and Hughes, 1991*); and (iii) a Δ*motAB* Δ*motCD* mutant (denoted as Δ*motABCD*), which has a flagellum (recognized by the pattern recognition receptors (PRRs)), but is non-motile due to the lack of motor proteins (*Toutain et al., 2005*) (see Key Resources Table). All bacterial strains displayed the expected phenotypes in scanning electron microscopy and motility assays (*Figure 1*).

### Reduced uptake of non-flagellated and non-motile *P. aeruginosa* mutants

A reduced uptake of non-flagellated and non-motile bacterial pathogens into bone-marrow- derived macrophages (BMDMs) has previously been reported (*Amiel et al., 2010*). Here, we employed macrophage invasion assays to test for the uptake of the aforementioned *P. aeruginosa* motility variants. We ensured comparable cell-to-cell contact of motile and non-motile bacterial cells by including a centrifugation step to establish bacterial contact with the adherent eukaryotic cells (*Figure 2—figure supplement 1A*). The phagocytic uptake of all three motility mutants was significantly reduced as compared to PA14 in primary BMDMs (*Figure 2A*), in RAW264.7 macrophages (*Figure 2B*) and in

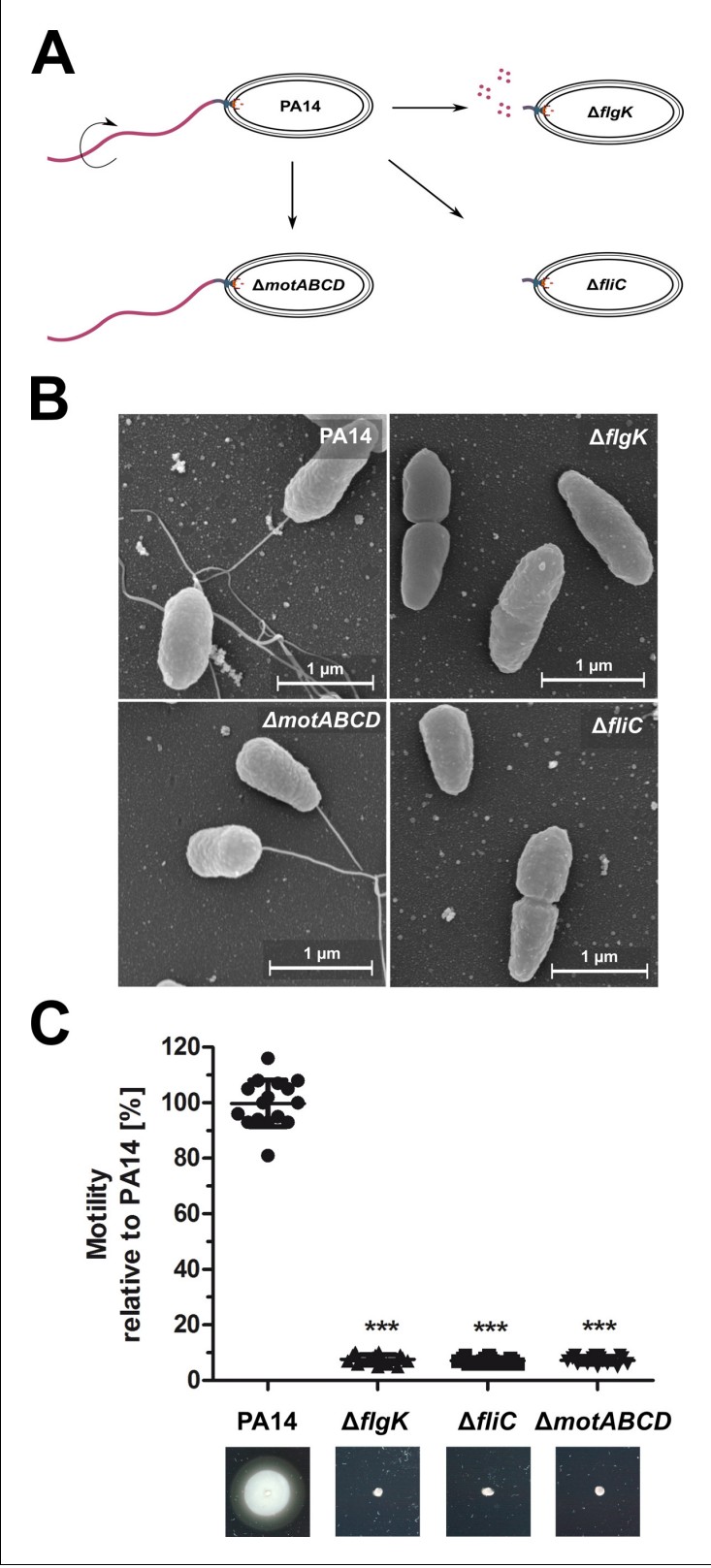

**Figure 1.** Phenotypic characterization of the *P. aeruginosa* strains. (**A**) Schematic depiction of the mono-flagellated, fully motile PA14 wild-type phenotype and three isogenic mutants which are non-motile due to the lack of flagella (Δ*fliC*), non-assembled flagellin (Δ*flgK*), or the lack of flagellar rotation (Δ*motABCD*). (**B**) Representative pictures of the individual *P. aeruginosa* motility variants analyzed by scanning electron microscopy.

*Figure 1 continued on next page*

*Figure 1 continued*

(**C**) Swimming motility assessed on semisolid agar after 16 hr. Mean ± standard deviation is displayed of 16 biological replicates from two independent experiments. \*\*\*p<0.001, one-way analysis of variance (ANOVA) (post-hoc test: Dunnett). See also: *Figure 1—source data 1*.

The online version of this article includes the following source data for figure 1:

**Source data 1.** Bacterial strains and plasmids used in this study.
**Source data 2.** Scanning Electron microscopy.
**Source data 3.** Motility Plates.

---

J774 cells (*Figure 2—figure supplement 1A* – 1C). Introduction of *fliC in trans* restored flagella production, motility, and phagocytic uptake in the *fliC* deletion mutant to wild-type levels (*Figure 2—figure supplement 2*).

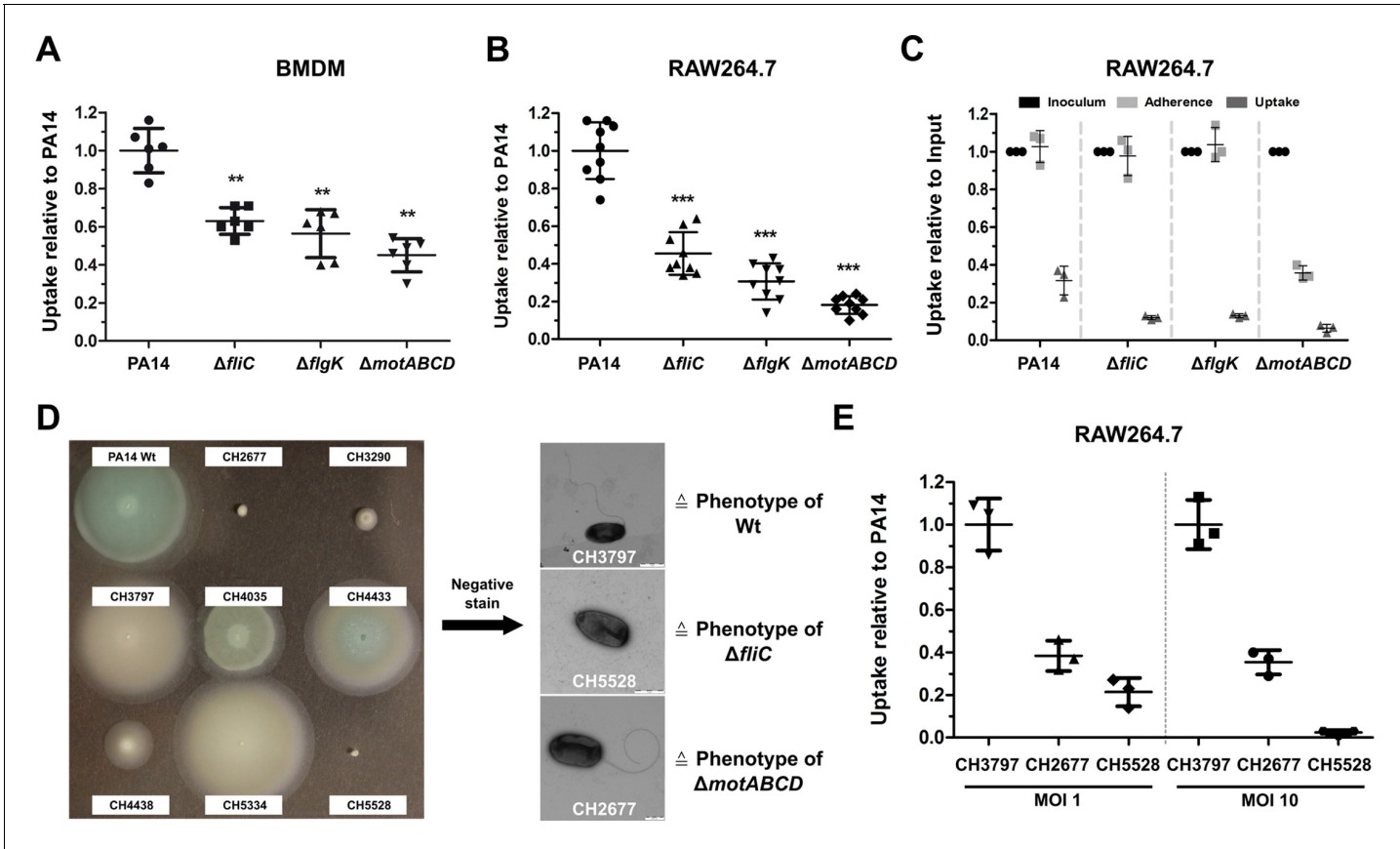

**Figure 2.** Reduced macrophage uptake of non-flagellated and non-motile *P. aeruginosa* strains. Phagocytic uptake 1 hr post infection of PA14 and the three motility variants into bone-marrow-derived macrophages (BMDMs) (**A**) and RAW264.7 cells (**B**) using a multiplicity of infection (MOI) of 1. The results of at least two independent experiments with three biological replicates are depicted. (**C**) Adherence of bacterial cells to RAW264.7 macrophages following treatment with 10 µM cytochalasin D to inhibit phagocytic uptake. The initial bacterial load was set to 100% (black, technical replicates) while the bacterial uptake without cytochalasin D is shown as a control. The data of three biological replicates is shown. (**D**) Left: Representative screen of clinical *P. aeruginosa* isolates for swimming motility on semi solid agar plates. Right: Analysis of clinical isolates using a negative-staining to identify the presence of flagella. (**E**) Phagocytic uptake of clinical isolates of *P. aeruginosa* in RAW264.7 cells 1 hr post infection using an MOI 1 and 10. Mean ± standard deviation of three biological replicates is displayed. \*\*p<0.01; \*\*\*p<0.001 in one-way analysis of variance (ANOVA) (post-hoc test: Dunnett).

The online version of this article includes the following source data and figure supplement(s) for figure 2:

**Source data 1.** Negative Stain.
**Figure supplement 1.** Phagocytosis and adherence of *P. aeruginosa* motility variants.
**Figure supplement 2.** FliC complementation restores the wild-type phenotype.

Addition of 10 µM cytochalasin D to the culture medium blocked bacterial uptake to less than 1% (*Figure 2—figure supplement 1D* – 1F). The adherence of the ΔfliC and ΔflgK strains to the cell surface of RAW264.7 macrophages and of J774 cells was comparable to that of the wild-type PA14 strain (*Figure 2C* and *Figure 2—figure supplement 1C*), indicating that the observed difference in bacterial uptake between the mutants and the wild-type is not due to impaired cell-to-cell contact. Of note, the flagellated, non-motile ΔmotABCD mutant adhered to the macrophages to a lesser extent, indicating that the presence of a non-functional flagella might disturb close contact between the bacteria and the host cells (*Laventie et al., 2019*).

To rule out the possibility that the lower uptake of the motility mutants can only be observed in the PA14 strain background, we screened our collection of clinical isolates for motility mutants (*Hornischer et al., 2019*) and selected one motile clinical isolate and two non-motile isolates (*Figure 2D*). As demonstrated by electron microscopy with negative-staining the motile isolate was flagellated, while one of the non-motile isolates lacked a flagellum, and the other harbored a flagellum, thus phenotypically recapitulating the ΔfliC and the ΔmotABCD mutant phenotypes. In line with the results for the PA14 strain, both non-motile clinical isolates were phagocytosed at lower rates by macrophages (*Figure 2E*) in comparison to the motile clinical isolate. Our results underscore the importance of a functional bacterial flagellum for phagocytic uptake.

## Dual-sequencing approach to analyze host–pathogen interactions

RNA sequencing of all three motility mutants in comparison to their respective wild-type under standard LB culture conditions was performed in order to rule out major shifts in the gene expression profiles among the strains. While the ΔmotABCD exhibited a differential expression of 206 genes as compared to PA14, the gene expression profile of the two other mutants, ΔfliC and ΔflgK, differed only in the expression of 49 and 47 genes, respectively (*Figure 3—figure supplement 1A*). This indicates that a non-functional flagellum does not have a large impact on the gene expression profiles of planktonic cells grown in rich medium conditions. The three motility mutants shared 20 genes, which were differentially expressed in comparison to PA14 (*Figure 3—figure supplement 1A*). These genes encode AlgZ, CheY, PctC and 17 hypothetical proteins.

To elucidate the specific crosstalk between *P. aeruginosa* and macrophages, we applied a Dual-sequencing approach. Bacteria were allowed to interact with RAW264.7 host cells (at a multiplicity of infection (MOI) of 1) for 3 hr prior to bacterial and eukaryotic RNA extraction and analysis (*Figure 3A*). In this experimental set-up, the ΔfliC and ΔflgK strains exhibited a similar profile of cell lysis as PA14 (see *Figure 3—figure supplement 2*). Furthermore, the pathogen-to-host mapping ratio of RNA sequencing data was comparable among infections with PA14 and the three motility mutants (*Figure 3—figure supplement 3*).

Eukaryotic transcriptional profiling revealed an activation of the RAW264.7 cells by *P. aeruginosa* with 647 genes induced (false discovery rate (FDR) $\leq$ 0.05) upon bacterial contact with PA14 compared to uninfected host cells. These included genes located in the TNF-, NF-κB-, and MAPK-signaling pathways, as well as those responsible for the expression of TLRs, and cytokine responses (*Figure 3B*). All three motility mutants induced host responses that were similar to those seen with PA14. Of the overall 1392 genes that were differentially expressed upon infection with either a motility mutant or PA14, 31.5% (438 genes) were shared by all strains (ΔfliC, ΔflgK, ΔmotABCD and PA14), while only 3.3% (46 genes) were exclusively regulated by the motility mutants and 5% (70 genes) exclusively by PA14 (*Figure 3—figure supplement 1B*).

In the following, we focused on the host genes that were regulated during infection with each of the mutants (ΔflgK, ΔfliC, and ΔmotABCD) compared to PA14. Therefore, the differentially expressed genes were sorted according to their FDR values. From the top 1000 genes with the lowest FDR values, 252 were commonly regulated upon infection with the three mutants and 330, 569, and 321 were exclusively regulated upon infection with the ΔflgK, ΔfliC, and ΔmotABCD mutant strain, respectively. A random permutation test (100000 repeats) revealed that only 8 ± 2.8 genes would have been expected to be commonly regulated by the three motility mutants by chance. This indicates that three motility mutants induce a similar specific transcriptional host response that is different from that of PA14.

Not only did the eukaryotic cells react to the presence of the bacteria, but bacterial gene expression profiles were also strongly affected by cell contact. A total of 2537 genes were differentially expressed in PA14 upon exposure to the eukaryotic cells when compared to planktonic in vitro

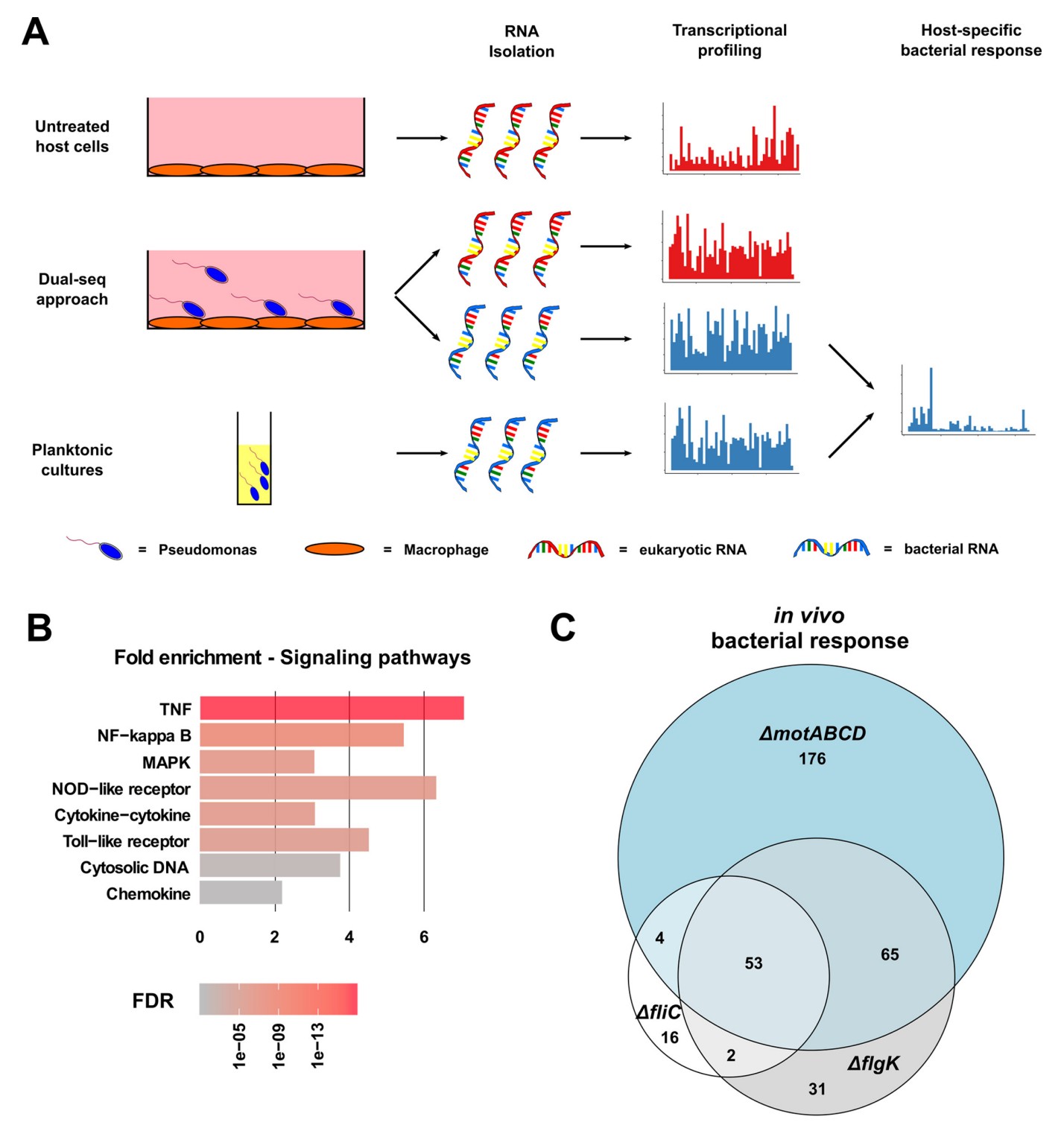

**Figure 3.** Dual-sequencing approach. (**A**) Experimental set-up to record the transcriptional profiles of infected RAW264.7 macrophages, *P. aeruginosa* upon host-cell contact, and *P. aeruginosa* grown in rich medium as a planktonic culture. (**B**) RAW264.7 pathways that are activated upon infection with PA14. (**C**) Venn diagram of the differentially expressed genes in the three motility mutants (relative to PA14) upon contact with the RAW264.7 macrophages as compared to planktonic conditions (FDR $\leq$ 0.05).

The online version of this article includes the following source data and figure supplement(s) for figure 3:

**Source data 1.** Summary of the 20 differentially expressed genes that are shared by the motility mutants in comparison to PA14 Wt.

*Figure 3 continued on next page*

*Figure 3 continued*

**Source data 2.** Significantly enriched host pathways.
**Figure supplement 1.** Venn Diagrams to visualize the dual-seq transcriptional responses.
**Figure supplement 2.** Lactate Dehydrogenase (LDH) assay of infected RAW264.7 macrophages.
**Figure supplement 3.** Pathogen-to-host read ratio.
**Figure supplement 4.** Functional GO term enrichment.
**Figure supplement 4—source data 1.** Enriched functions in the Pseudomonas variants.

growth conditions. Overall, 3875 genes were differentially expressed in any of the strains used in this study (Δ*fliC*, Δ*flgK*, Δ*motABCD* or PA14) upon host-cell contact (in vivo versus in vitro). 1888 (48.7%) of these genes were shared by all strains indicating a conserved transcriptional response to the presence of macrophages (*Figure 3—figure supplement 1C*).

Next, we identified genes that were specifically regulated in the three motility mutants upon contact to the host cells (*Figure 3C*). In the presence of phagocytes, 53 differentially regulated genes were commonly found in all three motility mutants when compared to PA14 upon host-cell contact. Of those 53 genes, 26 genes were also differentially expressed in the three motility mutants as compared to PA14 under planktonic LB conditions. Of these 26, 11 genes were upregulated and 15 were repressed exclusively in the motility mutants upon host-cell contact (*Figure 4*). Thus, these genes represent the host-regulated specific fraction of genes that were differentially expressed in all mutant strains when compared to PA14.

## Non-motile *P. aeruginosa* isolates do not activate the PIP₃ pathway, which results in decreased phagocytic uptake

We used the dual-sequencing approach to investigate whether the gene expression of the host after infection with the mutant strains differs from that upon infection with PA14. The RAW264.7 macrophage genes leading to either $PIP_2$ or $PIP_3$ activation were expressed at lower (albeit none significant) levels upon infection with the individual motility mutants in comparison to PA14 (*Figure 5A*). Nevertheless, respective functional host gene categories were enriched upon infection with the individual motility mutants as compared to PA14 (*Figure 5—source data 1*). Furthermore, in line with lower $PIP_3$ levels, we also found genes for the inositol 1,4,5-trisphosphate-sensitive calcium-release channel activity (ITPR1-3) to be lower expressed. Even with no significant enrichment for the single mutants, a lower expression of these genes in all three mutants is not random (FDR $\leq$ 0.05, random permutations in top 1000 genes of Δ*fliC*, Δ*flgK*, Δ*motABCD* mutants).

The activation of these pathways may be responsible for the observed enhanced phagocytic uptake of PA14 (*Figure 5A*). In fact, $PIP_3$ has been shown previously to induce Akt phosphorylation and to increase phagocytosis of the non-motile *P. aeruginosa* (*Demirdjian et al., 2018*). We therefore analyzed the role of the $PIP_3$ signaling pathway during phagocytosis of flagella mutants by supplementing histone-bound $PIP_3$ to RAW264.7 cells to complement the missing activation of the signaling cascade. The complementation of $PIP_3$ restored the uptake of the Δ*fliC* mutant to wild-type levels and significantly increased the uptake of Δ*flgK* and Δ*motABCD* strains (*Figure 5B*), confirming that the activation of the $PIP_3$ signaling pathway plays a central role in the phagocytosis of PA14. This activation is abrogated in *P. aeruginosa* mutants that are non-motile, and appears to be independent of flagellin production (Δ*fliC*), flagellum presence (Δ*flgK*) or flagellar functionality (Δ*motABCD*).

## Non-motile *P. aeruginosa* have altered expression levels of *arnT* and of genes involved in spermidine and pyoverdine biosynthesis

The putative spermidine synthesis operon, *PA14_63110*, *PA14_63120*, *PA14_63130*, as well as *arnT*, which encodes a 4-amino-4-deoxy-L-arabinose transferase responsible for masking the immunogenic 4'-$PO_4^{2-}$ group of the lipid A molecule, were expressed at significantly lower levels in the motility mutants in comparison to PA14 upon host-cell contact (*Figure 4*). Interestingly, the genes encoding proteins involved in spermidine synthesis were expressed at a higher level in PA14 upon host-cell contact as compared to its planktonic growth in rich medium conditions, whereas the spermidine synthesis genes were not differentially expressed (or were even expressed at lower levels) in the

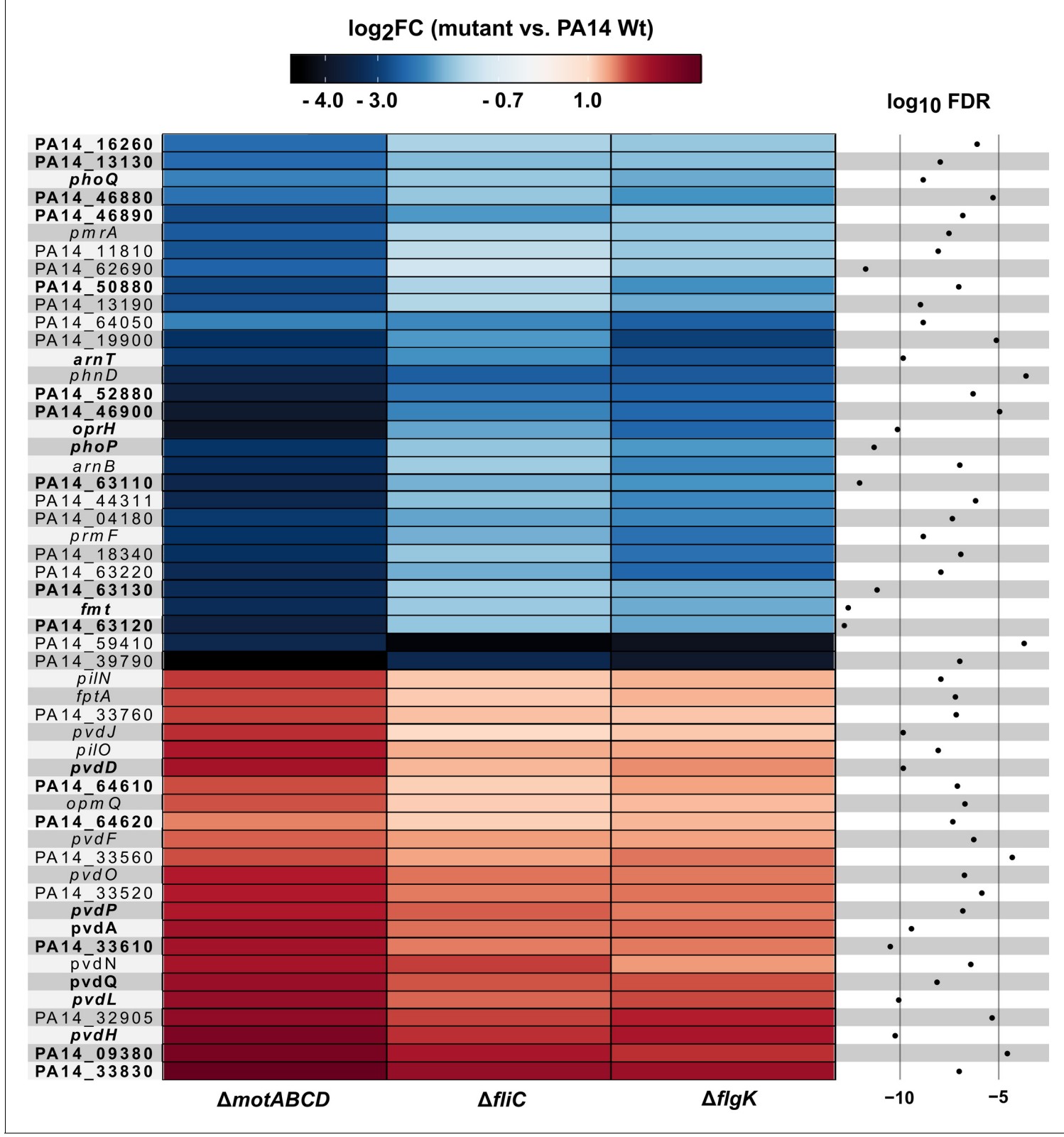

**Figure 4.** Motility mutant specific transcriptional responses in the presence of macrophages. Genes that were differentially expressed (log₂FC, glmTreat function, R package edgeR) in all three flagella mutants as compared to PA14 (n = 53 genes) are listed. The color code depicts the log₂FC expression from negative (blue) via 0 (white) to positive (red). Genes that are additionally differentially expressed between host-cell contact and LB growth

*Figure 4 continued on next page*

*Figure 4 continued*

conditions are shown in bold (FDR $\leq$ 0.05, 26 genes; additional FDR $\leq$ 0.0001 for analysis of variance (ANOVA)-like test using the function glmQLFTest of the edgeR package).

The online version of this article includes the following source data and figure supplement(s) for figure 4:

**Source data 1.** Bacterial gene regulation.
**Source data 2.** Positive interspecies correlation of 53 PA14 genes and their association with 74 host PIP$_3$ genes.
**Source data 3.** Negative interspecies correlation of 53 PA14 genes and their association with 74 host PIP$_3$ genes.
**Source data 4.** Differentially expressed genes between the *fliC* mutant and PA14 wild type after a medium switch from LB to DMEM medium containing FCS.

**Figure supplement 1.** Cytokine measurement in the supernatant of infected macrophages using ELISA.

three motility mutants under the same conditions. While *arnT* was not expressed at higher levels in PA14 upon host-cell contact, it was found to be expressed at lower levels in all motility mutants tested upon host-cell contact as compared to planktonic growth. Of note, switching the medium from LB to cell culture medium did not induce the same differential gene expression profile between $\Delta$*fliC* and PA14 indicating that host-cell contact but not the cell culture medium determines the differential *arnT* and spermidine biosynthesis gene expression profile between PA14 and its motility mutants (**Figure 3—figure supplement 3**).

Furthermore, genes involved in pyoverdine biosynthesis were significantly enriched among the more highly expressed genes in all motility mutants upon host-cell contact (**Figure 4**). Of those, *pvdL* (one of the most significantly differentially expressed gene), was upregulated in all strains upon host-cell contact. However, the increase in expression levels upon host-cell contact was significantly higher in the motility mutants than in PA14 as compared to respective planktonic growth. There was no differential expression of *pvdL* between the three motility mutants and PA14 under planktonic growth conditions in LB medium. Nevertheless, we found genes of the pyoverdine biosynthetic gene cluster (*pvdL* and *pvdQ*) to be differentially expressed between $\Delta$*fliC* and PA14 upon switching to cell culture medium (**Figure 3—figure supplement 3**). Therefore, the differential expression of pyoverdine in $\Delta$*fliC* versus PA14 cannot exclusively be attributed to the presence of the host cells.

To further elucidate the role of bacterial genes, which exhibited an altered expression in the motility mutants upon host-cell contact, particularly on the induction of the PIP$_3$ pathway, we used the data from the Dual-sequencing approach to perform a correlation analysis between the species. We correlated the 53 *P. aeruginosa* genes that were induced upon host-cell contact (**Figure 4**) with all expressed 12298 host genes (which had at least one count per million (cpm) in at least one sample). *PA14_63110* and *PA14_63120* from the spermidine synthesis operon and *PA14_18340*, *PA14_18350* (*arnA*), *PA14_18360*, and *PA14_18370* from the *arn* operon had a Spearman correlation coefficient $\geq$0.8 with 13% of the host genes, but with 24% of the PIP$_3$ genes, which is a significant enrichment (see **Figure 4—source data 2**). In total, the expression of 16 *P. aeruginosa* genes positively correlated ($\geq$0.8) with the expression of more PIP$_3$ associated genes than expected by chance (FDR $\leq$ 0.05). These genes included *pmrA* (PA14_63150), *phoQ* (PA14_49170) and *phoP* (PA14_49180).

The number of negative correlations of *P. aeruginosa* gene expression with the expression of PIP$_3$ host genes (rho $\leq$−0.8) was significantly enriched for *pvdH* (PA14_33500), *pvdD* (PA14_33650), and *pvdJ* (PA14_33630, **Figure 4—source data 3**). We also performed an interspecies correlation with data that shows the response of *P. aeruginosa* to the host (infected versus planktonic). Interestingly, expression of none of the *P. aeruginosa* genes correlated with a significant amount of PIP$_3$ genes (data not shown). We therefore hypothesize that bacterial spermidine synthesis is not induced by host PIP$_3$ genes, but that spermidine induces the expression of host PIP$_3$ genes.

## Non-motile *P. aeruginosa* isolates are more immunogenic

Activation of the *arn* operon has previously been linked to antibiotic sensitivity and lowered immunogenicity (**Breidenstein et al., 2011**; **Needham et al., 2013**). Screening of the supernatant of infected RAW264.7 and BMDM cells for the production of TNF-$\alpha$ and IL-6 revealed that the levels of these cytokines were significantly increased in all three motility mutants in comparison to PA14 (**Figure 4— figure supplement 1**), resembling the phenotype of a non-functional *arn* operon. This result

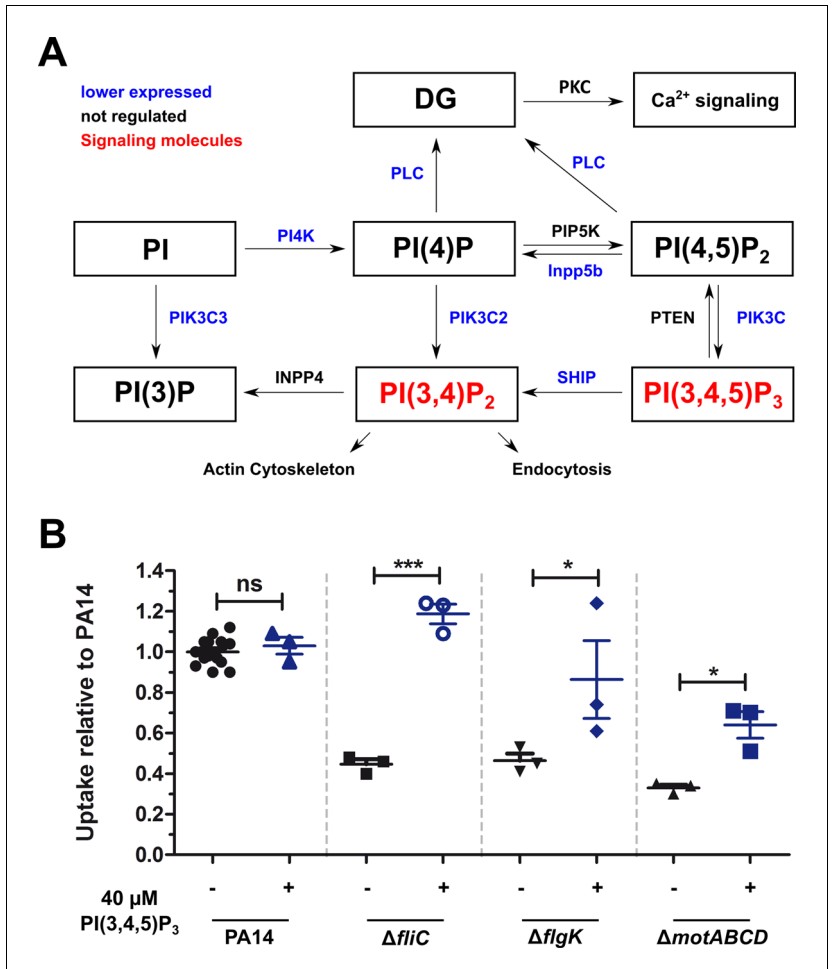

**Figure 5.** Non-motile variants of *P. aeruginosa* elicit a reduced activation of the PIP3 pathway in macrophages. (**A**) Dual-seq analysis revealed that the pathways leading to the signaling molecules PI(3,4)P$_2$ and PI(3,4,5)P$_3$ (in red), and the activation of the calcium-dependent cell rearrangements are expressed at lower levels in RAW264.7 cells infected with the flagellar mutants in comparison to PA14. The figure is based on annotations of the KEKK pathway mmu04070. The intersection of three pathways with each top 1000 genes of the motility mutants is shown. (**B**) Phagocytic uptake of PA14 and its motility variants in RAW264.7 cells 1 hr post infection using an MOI of 1 in the presence (blue, +) or absence (black, −) of 40 µM PI(3,4,5)P$_3$. Mean ± standard deviation of three individual experiments is displayed. *$p<0.05$; ***$p<0.001$ (one-way analysis of variance (ANOVA), Dunnett's post-hoc test).

The online version of this article includes the following source data for figure 5:

**Source data 1.** Enrichment of calcium and PIP$_3$-related functions of a downregulated (log$_2$FC < 0) subset of Top 1000 genes from Δ*fliC*, Δ*flgK* and Δ*motABCD* mutants.

demonstrates the potential role of *arnT* during host–pathogen interaction, suggesting that LPS modifications (i.e. phosphorylation status) modulate the immunogenicity of the individual motility mutants.

## Spermidine production is induced in flagellated *P. aeruginosa* upon host-cell contact to promote phagocytosis

We aimed to evaluate the impact of the presence of spermidine on phagocytic uptake (*Figure 6*). The minimal inhibitory concentration (MIC) of spermidine was determined to be approximately 10 mM for *P. aeruginosa* and did not differ between PA14 and the three motility mutants (data not shown). Exogenous addition of spermidine to the growth medium during infection experiments did not alter the phagocytic uptake of PA14 (*Figure 6—figure supplement 1A*). However, spermidine

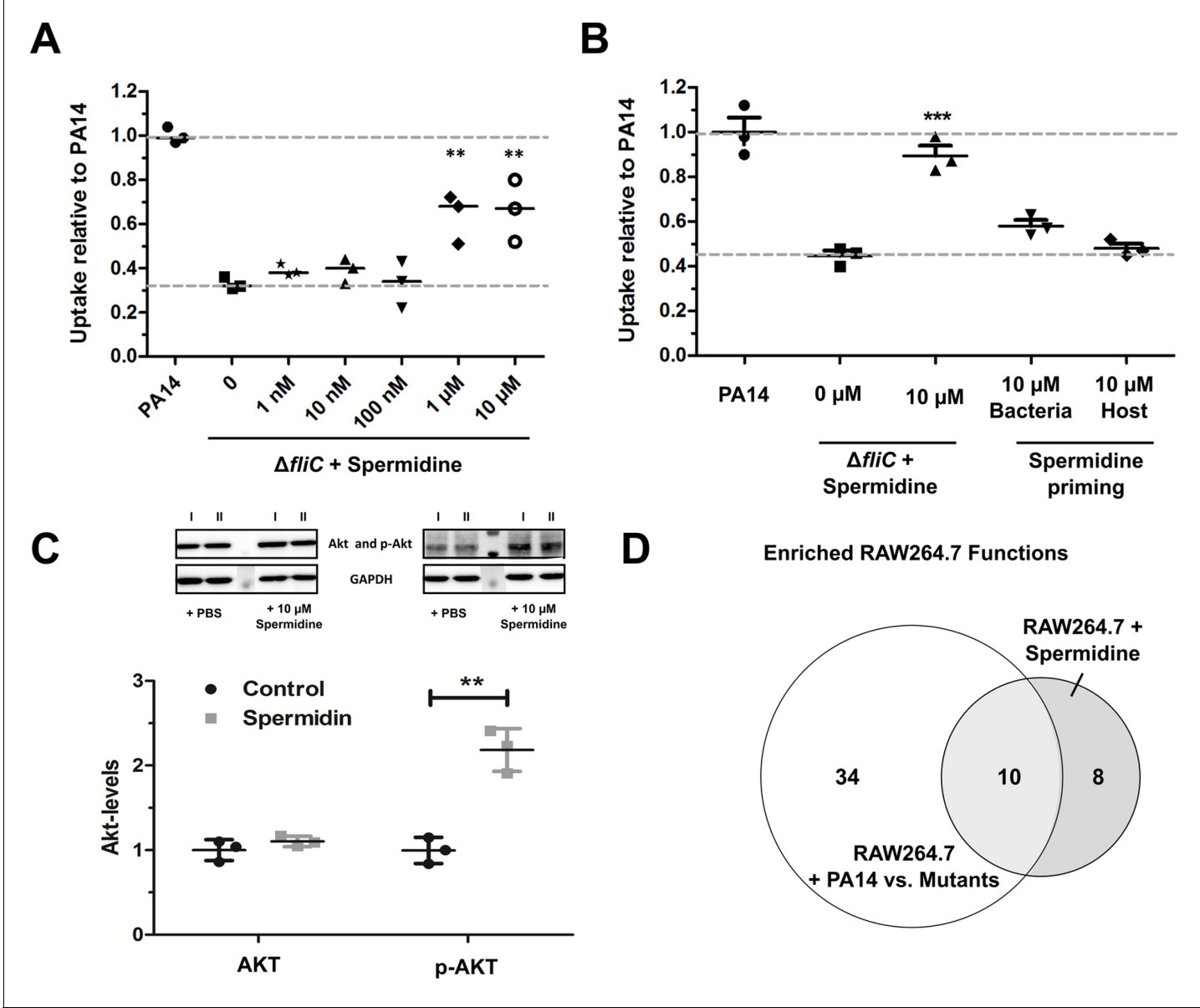

**Figure 6.** Spermidine enhances phagocytic uptake. (A) Spermidine was externally added at the indicated concentrations to the Δ*fliC* mutant. Using MOI of 1, the phagocytic uptake in RAW264.7 macrophages was examined 1 hr post infection. Mean ± standard deviation of three independent experiments are displayed. (B) Determination of phagocytic uptake after priming bacteria or RAW264.7 macrophages with 10 µM spermidine. Mean ± standard deviation of three representative biological replicates is displayed. ***p<0.001, two-way analysis of variance (ANOVA) (C) AKT phosphorylation was assessed and quantified by western blot following incubation of RAW264.7 macrophages for 1 hr with 10 µM spermidine. PBS treated cells served as control. N = 3. **p<0.01, two-way ANOVA (D) The transcriptional profile of spermidine-treated RAW264.7 macrophages was accessed and functional enrichment (of top 100 genes) was compared to the PA14-specific response (intersection of enriched functions from top 1000 genes of Δ*flgK*, Δ*motABCD*, and Δ*fliC* versus PA14 genes) of the macrophages. Venn diagram to illustrate the overlap of enriched (FDR ≤ 0.1) functions among genes that were differentially regulated following spermidine treatment and infection with PA14. The Venn diagram contains functional categories as a result of the functional enrichment using the DAVID tool with default categories (e.g. GO terms, COG ontology, and KEGG pathways). The online version of this article includes the following source data and figure supplement(s) for figure 6:

Source data 1. Spermidine induced gene induction.
Source data 2. Shared enriched functions of spermidine-treated and PA14-infected macrophages.
Figure supplement 1. Effect of spermidine on phagocytic uptake.

concentrations of 10 µM significantly increased the phagocytic uptake of the Δ*fliC* and Δ*flgK* mutants, while it did not have an effect on the uptake of the Δ*motABCD* strain (*Figure 6A* and *Figure 6—figure supplement 1B*). Of note, addition of spermidine to the bacteria during the infection experiments was crucial as sole priming of either bacteria or host cells prior to infection did not provide a significant effect (*Figure 6B*).

Since the motility mutants, which exhibit a low spermidine biosynthesis gene cluster expression, have previously been demonstrated to lack activation of the PIP$_3$ signaling pathway leading to Akt phosphorylation (*Hemmings and Restuccia, 2012*), we probed if spermidine is capable of altering Akt phosphorylation. We incubated RAW264.7 macrophages for 1 hr with 10 µM spermidine and quantified p-Akt using western blot. The addition of spermidine was uncovered to induce Akt phosphorylation, while Akt phosphorylation was not induced in untreated controls (*Figure 6C*).

To determine the global effect of purified spermidine on macrophages, transcriptional profiles of spermidine-treated RAW264.7 macrophages were produced and compared to the profiles of *P. aeruginosa* infected macrophages. We were not able to detect significant differential regulation of genes in the RAW264.7 cells upon treatment with spermidine. However, when we considered the top 1000 differentially regulated genes with the lowest FDR values, 34 genes were shared between the spermidine-induced macrophages and the *P. aeruginosa*-infected macrophages (22.8 genes expected by chance). We observed 10 functional gene categories that were enriched (FDR $\leq$ 0.1) in the spermidine-treated macrophages as well as in the PA14-infected macrophages. These 10 functional categories (*Figure 6—source data 2*) represent 55.6% of all enriched functions in the infected macrophages (*Figure 6D*) and thus indicates that spermidine indeed induces a host response that is similar to that observed upon PA14 infection.

Of note, genes encoding three CLK kinases (CLK1/3/4) were among the regulated genes in spermidine-treated RAW264 cells. Additional genes encoding serine/threonine kinases were significantly enriched (4-fold, FDR = 0.039) among the top 100 genes (IRAK2, PRKCZ, MAP4K2, MAP3K14, PIM3 in addition to CLK1/3/4). Importantly, serine/threonine kinases were also significantly enriched among the genes that were differentially regulated during infection with PA14. Altogether, the large overlap between the PA14- and the spermidine-induced transcriptional profile supports the hypothesis that spermidine plays a dominant role in the response of macrophages to PA14, which is largely phosphorylation dependent (*Figure 6—figure supplement 1C*).

To further confirm the role of spermidine, the operon was disrupted in PA14 through the deletion of *PA14_63120*. As measured by HPLC-MS, the Δ*PA14_63120* deletion mutant produced strongly reduced levels of membrane-associated spermidine as compared to PA14 (*Figure 7—figure supplement 1*). In line with our hypothesis that the induction of bacterial spermidine production triggers phagocytic uptake, the Δ*PA14_63120* mutant strain was significantly less phagocytosed by RAW264.7 cells in comparison to PA14. Phagocytic uptake of the Δ*PA14_63120* mutant by RAW264.7 macrophages could be restored to wild-type levels following the exogenous addition of spermidine (*Figure 7*). Of note, also the addition of norspermidine (*Bolard et al., 2019*) restored the phagocytic uptake of the non-spermidine producing mutants at comparable levels (*Figure 7* and *Figure 7—figure supplement 2*). Norspermidine is a triamine like spermidine but possesses a symmetric structure with two 3-aminopropyl chains while spermidine is asymmetric with one C4-chain (*Michael, 2016*). While spermidine can be found in prokaryotic and eukaryotic organisms, norspermidine appears to be restricted to prokaryotes and a role in antibiotic resistance and biofilm formation was previously discussed (*Qu et al., 2016*; *Tabor and Tabor, 1984*).

Altogether, our results demonstrate the role of (nor)spermidine in activating the PIP$_3$/Akt signaling pathway to induce phagocytosis of *P. aeruginosa*. Spermidine production is induced in *P. aeruginosa* upon host-cell contact promoting phagocytosis; however, this pathway is only induced in the presence of a functional flagellum.

## Discussion

In this study, we used a dual-sequencing approach in combination with an in vitro macrophage phagocytosis assay to assign a key role to bacterial motility in host–pathogen interactions. We found that motile bacteria respond differently to the presence of host cells and induce a gene expression program that determines the subsequent host response. This is a remarkable finding that indicates a dynamic conversation between the pathogen and the host. The pathogen is equipped with a set of

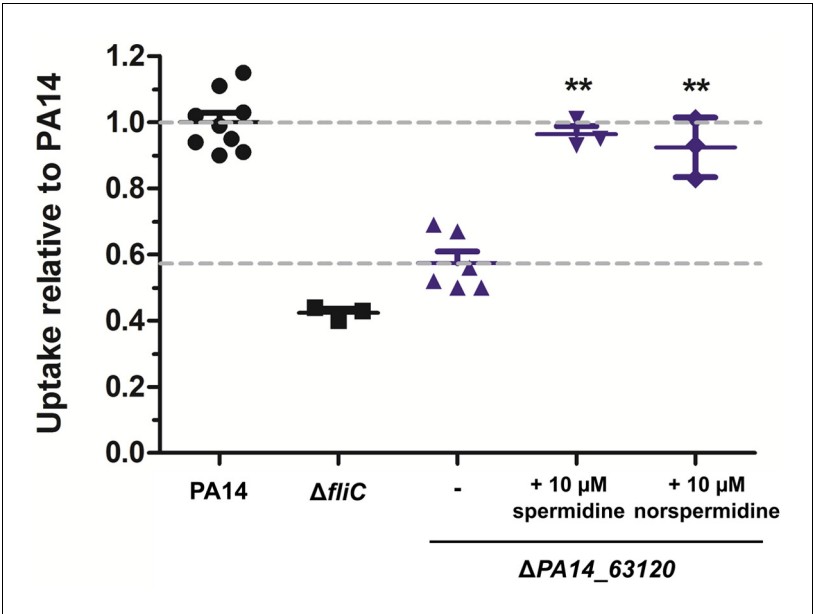

**Figure 7.** Spermidine and norspermidine restore phagocytic uptake of a spermidine knock-out mutant. Spermidine was externally added at the indicated concentrations to the $\Delta PA14\_63120$ mutant (blue). Using an MOI 1, the phagocytic uptake into RAW264.7 macrophages 1 hr post infection was examined and compared to PA14 and the $\Delta fliC$ control. Mean ± standard deviation is displayed. **$p<0.01$ (one-way analysis of variance (ANOVA), Dunnett's post-hoc test).

The online version of this article includes the following figure supplement(s) for figure 7:

**Figure supplement 1.** Quantification of spermidine using LC-MS.

**Figure supplement 2.** Effect of norspermidine on phagocytic uptake.

PAMPs such as the flagellum which, if present, is recognized by the host. However, as a direct response to the presence of host cells the pathogen expresses further traits adding a new level of complexity to the crosstalk between them (*Figure 8*).

Phagocytosis has been demonstrated to be differently triggered by bacterial motility variants (*Amiel et al., 2010*; *Lovewell et al., 2011*), and the search for a host-cell receptor to sense bacterial motility has been a subject of intense research. Here, we ruled out that differences in motility lead to altered adherence, and therefore to lowered phagocytosis rates. We show that the $\Delta fliC$ and $\Delta flgK$ mutants do not differentially adhere when centrifuged onto a macrophage cell layer. Only the $\Delta motABCD$ strain adhered at a lower rate to the eukaryotic cells, potentially due to non-functional *motAB* genes, which have been implicated to play a role in a pili-dependent adherence (*Laventie et al., 2019*). It seems that differences in motility, rather than in adherence, impact phagocytic uptake. As previously demonstrated by Demirdjian et al., the defect in the phagocytosis of non-motile variants could be restored by adding $PIP_3$ to the host cells (*Demirdjian et al., 2018*). $PIP_3$ is a membrane-associated phospholipid that acts as a signaling molecule for many cellular processes such as the Akt-pathway (*Czech, 2000*; *Traynor-Kaplan et al., 1988*). Our dual-sequencing approach revealed that calcium transport systems and the signaling pathway leading to the production of $PIP_2$ and $PIP_3$ were activated in the host cell upon infection with PA14. Both pathways are known to be involved in phagocytosis, and a lack of their activation in the non-motile strains could explain the lower phagocytosis rates (*Booth et al., 2001*; *Nunes and Demaurex, 2010*).

Our findings suggest that the non-motile variants are lacking a factor to stimulate $PIP_3$ activation and calcium transport. The transcriptional profiling of *P. aeruginosa* revealed that strains with impaired motility or lack of flagella exhibit altered expression of genes involved in the biosynthesis of bacterial surface structures. Of note, the motility variants differentially expressed genes encoding these structures exclusively upon host-cell contact and not under standard lab (LB or BM2 growth) conditions.

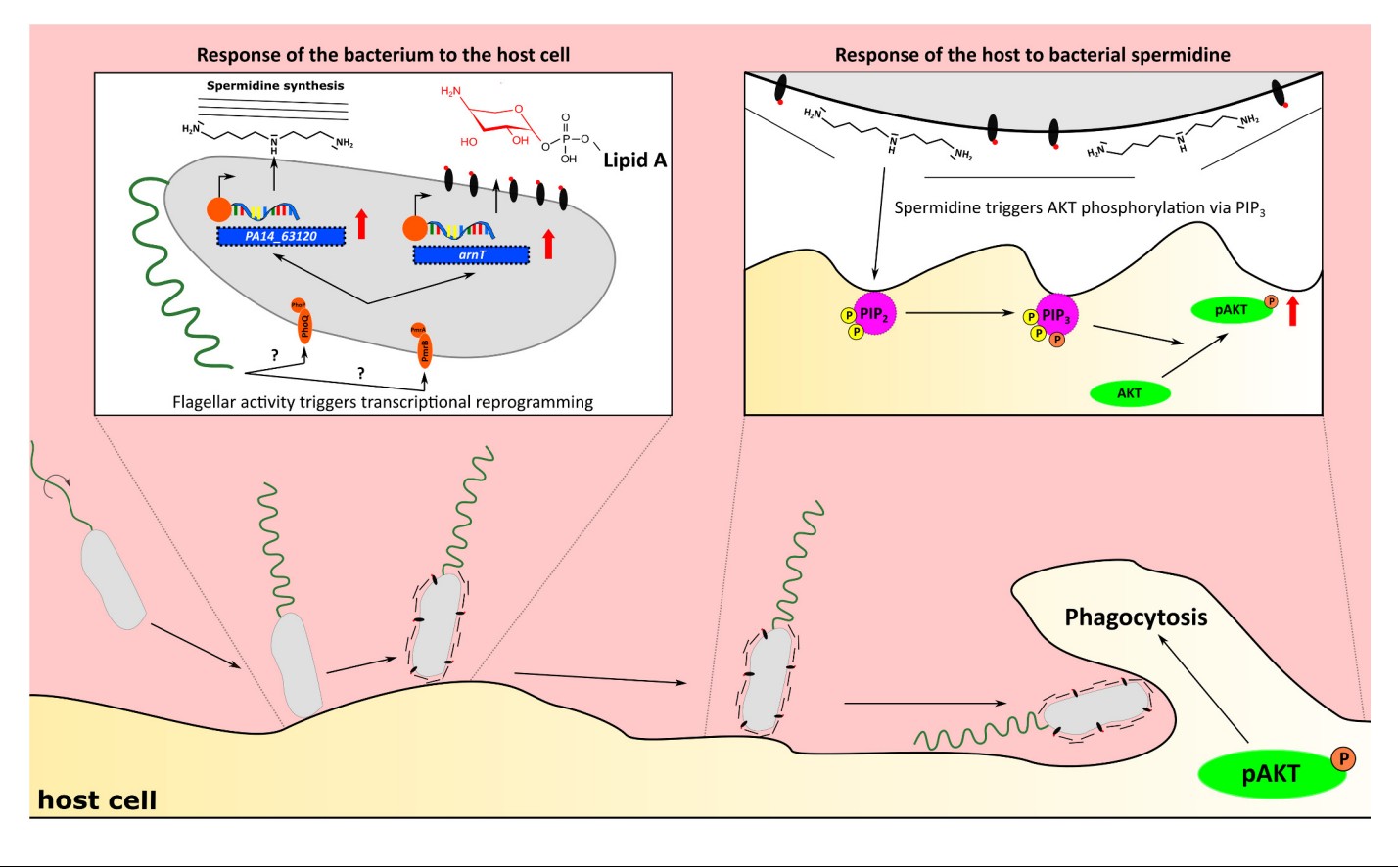

**Figure 8.** Flagellum-dependent *Pseudomonas*–macrophage interaction. Upon host contact, changes in flagellar activity (possibly sensed by PmrAB and PhoPQ) trigger a transcriptional program, which results in the synthesis of membrane-associated spermidine and the masking of the Lipid A. In the host, AKT phosphorylation through PIP$_3$ is stimulated by spermidine to increase phagocytosis.

A membrane-associated spermidine biosynthesis operon, which has been previously connected to antibiotic resistance (*Johnson et al., 2012*), was expressed at significantly lower levels in all motility mutant strains compared to PA14 wild-type upon host-cell contact. We furthermore demonstrate that externally added spermidine was able to activate the Akt-pathway, restoring the level of phagocytosis of the flagellar mutants and of a *P. aeruginosa* strain incapable of producing spermidine (ΔPA14_63120). We show that spermidine itself induced a transcriptional pattern in the host cell, which was similar to that of a PA14-infected host cell and involves triggering of global phosphorylation cascades (*Bennetzen et al., 2012*). In particular, serine/threonine kinases, which were enriched in spermidine-treated cells, are known to phosphorylate Akt among other proteins (*Burgering and Coffer, 1995*). Spermidine as a polyamine has been shown to mimic the effect of Ca$^{2+}$ activation (*Koenig et al., 1983*; *Toninello et al., 2004*), modulate immune responses (*Latour et al., 2020*; *Theoharides, 1980*) and trigger phagocytosis (*TranVan Nhieu et al., 2004*). Furthermore, polyamine depletion has previously been shown to result in increased expression of LPS-induced proinflammatory genes, coinciding with our findings of an enhanced production of TNF-α and IL-6 in the motility mutants (*Van den Bossche et al., 2012*). Enhanced expression of proinflammatory cytokines in our motility mutants may additionally be explained by the differential expression of *arnT*, which was significantly downregulated in all non-motile *P. aeruginosa* strains upon host-cell contact (*Felgner et al., 2016*; *Needham et al., 2013*). ArnT is a 4-amino-4-deoxy-L-arabinose transferase that masks the 4'-phosphate group of the lipid A molecule to avoid immune recognition via TLR-4 (*Maeshima and Fernandez, 2013*).

Interestingly, both the production of spermidine, and the expression of ArnT are governed by the two-component system PmrAB (*Gooderham et al., 2009*; *McPhee et al., 2006*) which is located on the same operon as *PA14_63120*. *PmrAB* was downregulated upon host-cell contact in the mutant strains to the same extent as *arnT* and the spermidine biosynthesis operon. In *E. coli*, the PmrAB homolog is known to regulate motility genes (*Sperandio et al., 2002*), and the *Legionella pneumophila* PmrAB appears to fine-tune flagellar gene expression in a negative feedback loop (*Al-Khodor et al., 2009*). Upon host-cell contact PmrAB as well as a second two-component regulatory system, PhoPQ, which regulates the *arn* operon (*Gooderham et al., 2009*) and *PA14_63120* (*Bolard et al., 2019*), were affected in the motility mutants. *PhoQ*, is regulated by the PmrAB two-component regulatory system (*McPhee et al., 2003*) and was previously shown to affect twitching motility, biofilm formation, attachment and cytotoxicity in human lung epithelial cells (*Gooderham et al., 2009*). It seems conceivable that PmrAB and PhoQP are induced by shared triggers upon host-cell contact. In *Salmonella*, PhoQ is activated within acidified phagosomes and host tissues (*Prost and Miller, 2008*) and both two-component systems are activated by $Ca^{2+}$ and $Mg^{2+}$ limiting conditions (*Guragain et al., 2016*; *McPhee et al., 2003*). This is interesting as it implicates that the motile bacterial cell may detect changes in the ionic environment, leading to the activation of these two-component systems. Since their activation is dependent on motility and the presence of a host cell, one might speculate that if a motile bacterium hits a cell, changes in bacterial motility in response to the physical barrier may entail altered ion fluxes across the bacterial cell membrane. Changes in pericellular ion homeostasis might serve as a stimulus to activate the two-component systems and the production of polyamines, which are subsequently perceived by the host cell, thereby inducing phagocytosis. Our results highlight that the combined use of cell culture assays with a dual-sequencing approach has the potential to uncover as yet unappreciated dynamic molecular crosstalk between motile *P. aeruginosa* isolates and the eukaryotic host. A deeper understanding of host–pathogen interactions on a molecular level will be essential for the identification of new therapeutic targets as the basis to address the important medical need of counteracting bacterial infections.

# Materials and methods

## Key resources table

| Reagent type (species) or resource | Designation | Source or reference | Identifiers | Additional information |
|---|---|---|---|---|
| Chemical compound, drug | Bacto-Agar | BD | 214010 | - |
| Chemical compound, drug | Gentamycin | Sigma | G1397-10ml | - |
| Chemical compound, drug | PtdIns(3,4,5)P$_3$ | Echelon | P-3916 | - |
| Chemical compound, drug | Shuttle PIPTM Carrier 2 | Echelon | P-9C2 | - |
| Chemical compound, drug | DMEM | Gibco | 31885–023 | - |
| Chemical compound, drug | RPMI | Gibco | 21875–034 | - |
| Chemical compound, drug | FCS | Biowest | S1810-500 | - |
| Chemical compound, drug | BSA | Sigma | A9418-50 | - |

*Continued on next page*

*Continued*

| Reagent type (species) or resource | Designation | Source or reference | Identifiers | Additional information |
|---|---|---|---|---|
| Chemical compound, drug | HEPES | Gibco | 15630–056 | - |
| Chemical compound, drug | Cytochalasin D | Sigma | C2618-200UL | - |
| Chemical compound, drug | Spermidine | Sigma | S0266-25G | - |
| Chemical compound, drug | Norspermidine | Sigma | I1006A-100G-A | - |
| Chemical compound, drug | TritonX-100 | BioRad | #161–0407 | - |
| Chemical compound, drug | RNAprotect Bacteria Reagent | Qiagen | #76506 | - |
| Chemical compound, drug | Pyridine, HPLC Grade, 99.5+% | Alfa Aesar | 110-86-1 | - |
| Chemical compound, drug | Phenylisothiocyanate (PITC) | Sigma | 78780 | - |
| Antibody | Rabbit anti-AKT | Cell Signaling Technology | #9272 | 1:1000 |
| Antibody | Rabbit anti-p-AKT (Ser473) | Cell Signaling Technology | #9271 | 1:1000 |
| Antibody | Mouse anti-GAPDH | Calbiochem | CB1001 | 1:10000 |
| Antibody | Goat anti-mouse IgG+IgM (H+L) | Dianova | 115-035-068 | 1:10000 |
| Antibody | Goat anti-rabbit IgG (H+L) | Dianova | 111-035-045 | 1:10000 |
| Commercial assay, kit | NEBNext Single Cell/low input Input RNA Kit | NEB | E6240S | - |
| Commercial assay, kit | Ribo-Zero rRNA removal Kit | Illumina | - | - |
| Commercial assay, kit | NEBNext Ultra II Directional RNA Library Prep Kit | NEB | E7760L | - |
| Commercial assay, kit | ELISA MAX Standard Set Mouse TNF-α | BioLegend | #430901 | - |
| Commercial assay, kit | ELISA MAX Standard Set Mouse IL-6 ELISA MAX Standard Set Mouse IL-6 ELISA MAX Standard Set Mouse IL-6 | BioLegend | #431301 | - |

*Continued on next page*

*Continued*

| Reagent type (species) or resource | Designation | Source or reference | Identifiers | Additional information |
|---|---|---|---|---|
| Cell line (*M. musculus*) | RAW264.7 macrophages | - | RRID:CVCL_0493 | - |
| Cell line (*M. musculus*) | J774 cells | - | RRID:CVCL_0358 | |
| Cell line (*M. musculus*) | BMDMs C57BL/6 | Own breeding | N/A | - |
| Cell line (*M. musculus*) | L929 | - | RRID:CVCL_0462 | - |
| Strain, strain background (*P. aeruginosa*) | PA14 wild-type | *Amiel et al., 2010* | N/A | - |
| Strain, strain background (*P. aeruginosa*) | ΔfliC | *Amiel et al., 2010* | N/A | - |
| Strain, strain background (*P. aeruginosa*) | ΔflgK | *Amiel et al., 2010* | N/A | - |
| Strain, strain background (*P. aeruginosa*) | ΔmotABCD | *Amiel et al., 2010* | N/A | - |
| Strain, strain background (*P. aeruginosa*) | ΔPA14_63120 | This study | N/A | - |
| Strain, strain background (*P. aeruginosa*) wild-type | CH2677 | *Hornischer et al., 2019* | N/A | Clinical Isolate |
| Strain, strain background (*P. aeruginosa*) wild-type | CH3290 | *Hornischer et al., 2019* | N/A | Clinical Isolate |
| Strain, strain background (*P. aeruginosa*) | CH3797 | *Hornischer et al., 2019* | N/A | Clinical Isolate |
| Strain, strain background (*P. aeruginosa*) | CH4035 | *Hornischer et al., 2019* | N/A | Clinical Isolate |
| Strain, strain background (*P. aeruginosa*) | CH4433 | *Hornischer et al., 2019* | N/A | Clinical Isolate |
| Strain, strain background (*P. aeruginosa*) | CH4438 | *Hornischer et al., 2019* | N/A | Clinical Isolate |
| Strain, strain background (*P. aeruginosa*) | CH5334 | *Hornischer et al., 2019* | N/A | Clinical Isolate |
| Strain, strain background (*P. aeruginosa*) | CH5528 | *Hornischer et al., 2019* | N/A | Clinical Isolate |
| Strain, strain background (*M. musculus*) | C57BL/6 | Own Breeding | N/A | - |
| Recombinant DNA reagent | pEX18-Gm | Lab stock | N/A | Suicide vector |
| Software, algorithm | ImageJ | - | RRID:SCR_003070 | v1.52p |

*Continued on next page*

*Continued*

| Reagent type (species) or resource | Designation | Source or reference | Identifiers | Additional information |
|---|---|---|---|---|
| Software, algorithm | Prism | Graphpad | N/A | v5.01 |
| Software, algorithm | Zeiss SEM Smart | Zeiss | N/A | v5.05 |
| Software, algorithm | ITEM Build 1210 | Olympus Soft Imaging | N/A | - |
| Software, algorithm | Skyline | MacCoss Laboratory | N/A | v18.305 |
| Software, algorithm | R Project | - | RRID:SCR_001905 | v3.6.1 |
| Software, algorithm | Analyst | ABSciex | N/A | v1.6.2 |
| Software, algorithm | Fastqc | N/A | RRID:SCR_014583 | v0.11.4 |
| Software, algorithm | tophat2 | N/A | RRID:SCR_013035 | v2.0.12 |
| Software, algorithm | Bowtie2 | N/A | RRID:SCR_005476 | v2.3.4.1 |
| Software, algorithm | Samtools | N/A | RRID:SCR_002105 | v0.1.19.0 |
| Software, algorithm | bedtools | N/A | RRID:SCR_006646 | v2.29.0 |
| Software, algorithm | DAVID (https://david.ncifcrf.gov/) | N/A | RRID:SCR_001881 | v6.7 |
| Software, algorithm | R library Rsubread | N/A | RRID:SCR_016945 | v1.34.7 |
| Software, algorithm | R library edgeR | N/A | RRID:SCR_012802 | v3.24.3 |
| Software, algorithm | R library eulerr | *Larsson, 2018* | N/A | v6.0.0 |
| Software, algorithm | R library superheat | *Barter and Yu, 2018* | N/A | v0.1.0 |
| Software, algorithm | R library pheatmap | N/A | RRID:SCR_016418 | v1.0.12 |
| Software, algorithm | R library ggplot2 | N/A | RRID:SCR_014601 | v3.1.1 |
| Software, algorithm | R library RColorBrewer | N/A | RRID:SCR_016697 | v1.1–2 |
| Other | Reference genome *M. musculus* | GRCm38/mm10 | N/A | - |
| Other | Reference genome *P. aeruginosa* UCBPP-PA14 | NC_008463.1 | N/A | - |

## Mice

All animal experiments were performed according to guidelines of the German Law for Animal Protection and with permission of the local ethics committee and the local authority LAVES (Niedersächsisches Landesamt für Verbraucherschutz und Lebensmittelsicherheit). C57BL/6 mice derived from our own breeding.

## Strains and preparation of inoculum

Bacterial strains are shown in *Figure 1—source data 1*. Strain construction utilized the suicide vector backbone pEX18-Gm. Pseudomonas strains were grown in LB medium (with 7.5 g/l NaCl) at 37°C. Overnight cultures were sub-cultured and grown for 3.5 hr to mid-log phase. Following growth, bacteria were washed twice and adjusted to the desired $OD_{600}$ in pyrogen-free PBS. Plating served as control. The vector pSEVA2513 (*oriT*, $P_{EM7}$, Km$^R$) was used to complement *fliC* in the deletion strain (*Durante-Rodríguez et al., 2014*; *Silva-Rocha et al., 2013*). 500 µg/ml kanamycin was supplemented to the cultures to maintain the plasmid in the *P. aeruginosa* strains.

## Motility assay

The motility of the Pseudomonas strains was assessed on semisolid BM2-agar plates containing 0.3% (wt/vol) agar. 2 µl of a bacterial overnight culture was stabbed into the agar, and the culture's diameter was assessed after 18 hr as measure of motility.

## Negative staining

Overnight cultures were fixed in 2% glutaraldehyde and negatively-stained with 2% uranyl acetate. Samples were examined in a Zeiss TEM 910 at 80 kV with calibrated magnifications. Images were recorded with a Slow-Scan CCD-Camera (ProScan, 1024 × 1024) and ITEM-Software (Olympus Soft Imaging Solutions).

## Scanning electron microscopy

Bacteria were cultured overnight prior to fixation in glutaraldehyde (2% final) and stored at 4°C. After washing with TE buffer (20 mM TRIS, 1 mM EDTA, pH 6.9) bacteria were placed onto poly-L-lysine covered cover slips, dehydrated with a graded series of acetone, then critical point dried with $CO_2$ and sputter coated with gold palladium. Samples were imaged with a Zeiss Merlin field emission scanning electron microscope (FESEM) at an acceleration voltage of 5 kV using the Everhart-Thornley SE-detector and the Inlens SE-detector in a 25:75 ratio. Images were recorded with the ZEISS SEM software version 5.05.

## Western blot

RAW264.7 macrophages were incubated in the presence or absence of 10 µM spermidine. After 1 hr, the cells were washed twice and resuspended in 8x SDS loading buffer. The proteins were separated using a 10% SDS-PAGE at 100 V for 1 hr. The membrane was dried and blocked for 1 hr using 10% heat-inactivated FCS in TBST-T. Primary staining was accomplished using Akt antibody #9272 (1:1000, Cell signaling), p-Akt antibody #9271 (1:1000, Cell signaling) and anti-GAPDH mouse antibody (1:10000, Merck). Secondary staining was done with 115-035-068 goat IgG anti-mouse IgG-IgM (H+L)-HRPO (1:10000, Dianova) and 111-035-045 goat IgG anti-rabbit IgG (H+L)-HRPO (1:10000, Dianova). The blot was developed using Lumi-Light Western Blotting Substrate (Roche) according to the manufacture's protocol. The band intensities were measured using FIJI and normalized to the GAPDH control band.

## Growth curves

Bacteria were grown overnight at 37°C and adjusted to $OD_{600}$ of 0.001 in LB. 200 µl were pipetted in a honeycomb multiwall plate. For measurement of growth, the honeycomb plate was incubated for the indicated time and $OD_{600}$ was determined in a multi-well reader (Bioscreen). $OD_{600}$ was measured every 15 min. After measurement, samples were blank corrected.

## Cell lines

RAW264.7 (ATCC number: TIB-71TM) and J774 (ATCC number: TIB-67) cells were cultured in DMEM (4.5 g/l Glucose) and supplemented with 10% FCS, 20 mM HEPES and 1% BSA at 37°C, 5% $CO_2$. BMDMs were isolated from the bones of C57BL/6 mice and cultured in RPMI supplemented with 20% (v/v) conditioned L929 (ATCC number: CCL-1) medium to allow for differentiation into macrophages for at least 7 days. All cell cultures used in this study were tested and confirmed as mycoplasma-free.

## Adherence assays

Eukaryotic cells were seeded in DMEM at a concentration of $5 \times 10^5$ cells in a 24-well plate for the phagocytic uptake. Overnight cultures were sub-cultured and grown for 3.5 hr to mid-log phase. Following growth, bacteria were washed twice with PBS and adjusted to MOI 1 ($5 \times 10^5$ bacteria per 100 µl) in DMEM. Cytochalasin D was added to cell cultures at the indicated concentrations 30 min prior to infection to block phagocytosis. Upon infection, the bacteria were centrifuged at 1000 x g for 5 min to ensure cell-to-cell contact. After 1 hr, the supernatant was removed. The cells were then gently washed twice with PBS and lysed using PBS containing 1% (v/v) TritonX-100. Serial dilutions were used to determine the adherent bacteria by plating.

## Invasion assays

Eukaryotic cells were seeded in DMEM at a concentration of $5 \times 10^5$ cells in a 24-well plate for phagocytic uptake. The assay was performed as described before using MOIs of 1 and 10 and adjusted as described above in DMEM (*Gahring et al., 1990*). A centrifugation step at 1000 x g for 5 min was used to ensure cell-to-cell contact. At 2 hr, cells were washed with PBS and lysed with PBS containing 1% (v/v) TritonX-100. CFUs were determined by plating of serial dilutions and compared to the corresponding parental strains. Spermidine was diluted in DMEM without FCS and added at the indicated time points and concentrations. 400 µM $PIP_3$-histone complexes were constructed as described previously (*Demirdjian et al., 2018*), diluted 1:10 in DMEM and loaded onto the cells 10 min prior to infection.

## ELISA measurement

Macrophage supernatants were taken 6 hr post infection. The TNF-$\alpha$ and IL-6 ELISA Max Standard Kit (Biolegend) were used to determine the TNF-$\alpha$ and IL-6 levels according to the manufacturer's manual. Three different biological replicates were analyzed and a PBS-treated group served as negative control.

## LDH measurement

RAW macrophages were infected with $5 \times 10^5$ or $5 \times 10^6$ bacteria of the respective strain in phenol-red free DMEM media as described above. The supernatant was collected 3 hr and 5 hr post-infected and LDH levels were measured using the CytoTox 96 Non-radioactive Cytotoxicity Assay (Promega) according to the manufacture's protocol. PBS serves as negative, viability control and 10% (v/v) Triton-X100 as killing control.

## Spermidine quantification

Mid-log-phase cultures were grown in BM2-Medium. Cells were collected by centrifugation (5000 x g, 10 min, RT) and spermidine was extracted using 1 M NaCl in 10 mM HEPES as previously described (*Johnson et al., 2012*). For derivatization, 100 µl sample were mixed with 150 µl of derivatization solution (47.5% EtOH; 46.5% Pyridine (Alfa Aesar); 7% PITC (Sigma-Aldrich)) and incubated for 60 min at room temperature with 600 rpm. The samples were centrifuged for 5 min at 1000 rpm and 50 µl of the supernatant was transferred into an LC–MS-vial and immediately measured. A volume of 5 µl per sample was injected into the LC–MS system. The LC–MS measurements were done with an ABSciex QTRAP 6500 mass spectrometer, equipped with an IonDrive Turbo-V ion source. A 1290 Agilent UHPLC-system with binary pump, thermostated autosampler and column oven were used for the LC separation. A XSelect CSH Phenyl-Hexyl column (Waters; 4.6 × 50 mm, 2.5 µm pore size) were used at 30˚C. The flowrate was set to 600 µl per minute. Solvent A was water with 0.1% formic acid, solvent B acetonitrile with 0.1% formic acid. The gradient started with 1% B, was held at 1% B until 0.1 min and then increased to 100% B within 4 min. The composition with 100% B was held for 30 s, followed by a return to the 1% B starting condition. The overall run time was 7 min.

The PITC derivatives were measured in positive MRM mode. Quantifier for PITC-Spermidine was the transition from 551.1 to 416.2 Da, and the transition from 551.1 to 193.0 Da was used as qualifier. As general parameters for the MS the following values were used: curtain gas 20 psi nitrogen; ion spray voltage 5500 V; temperature 400˚C; gas1 45 psi; gas2 60 psi (both compressed air); entrance potential 10 V. The quantification of the MRM data was done with the software Skyline (MacCoss Laboratory, University of Washington).

## RNA isolation and RNA sequencing

Planktonic cultures: Subcultures of Pseudomonas strains were grown to OD = 2, cell pellets were subsequently treated with RNAprotect and the RNA isolated as previously described (*Kordes et al., 2019*).

Dual-sequencing: RAW264.7 cells were infected with an MOI of 1 with the respective strains. As control, PA14 and Δ*fliC* were grown in LB for 3.5 hr and adjusted to the infection dose of $5*10^6$ bacteria/ml in order to keep the experimental conditions similar to that of the infection experiments. The adjusted bacteria were washed and transferred into cell culture medium. After 3 hr, RNA of all samples was extracted using RNAprotect and isolated as previously described (*Kordes et al., 2019*). Ribosomal RNA was depleted using the RiboZero Gold Kit (Qiagen, Venlo, Netherlands) and ERCC RNA Spike-In control mixes (Ambion Thermo Fisher Scientific Inc, Waltham, Massachusetts, USA) were added to the RNA samples. Libraries were prepared with the NEBNext Ultra II Directional RNA kit (New England Biolabs, Ipswich, Massachusetts, USA). For sequencing of BMDMs after the addition of spermidine, the NEB SingleCell Low Input Kit (unstranded) was used. For all samples, 50 nt paired final readings were sequenced on the Illumina NovaSeq 6000 (Illumina, San Diego, California, USA).

## Data analyses – dual sequencing

Reads of eukaryotic RNA were mapped to the mm10 (UCSC) mouse genome using tophat2 and gene reads were extracted using the R package Rsubread (*Liao et al., 2019*). Sequencing reads of bacterial RNA were mapped to the PA14 genome NC_008463.1 with bowtie2 and reads per gene were extracted with bedtools intersect with a minimal gene overlap of 50%. Parallelization was done using the command line tool parallel (*Tange, 2011*).

Subsequent analyses were done with R version 3.6.1. Analysis of differential gene expression was performed using the R package edgeR version 3.24.3 (*McCarthy et al., 2012*; *Robinson et al., 2010*). For eukaryotic RNA, normalization factors were calculated using ERCC RNA Spike-In controls. To filter for genes that are considered as expressed, we kept only genes that have more than one normalized read (count per million (cpm)) in at least n samples (n: number of replicates; infected/PBS treated cells = 2, spermidine-treated cells = 3). These 11129 (infected and PBS treated cells) and 10994 genes (spermidine-treated cells), respectively, were used to calculate differential gene expression applying Genewise Negative Binomial Generalized Linear Models with Quasi-likelihood Tests (edgeR packages glmQLFit and glmQLFTest). Functional enrichment was performed with DAVID (https://david.ncifcrf.gov/) with fold enrichment and FDR added as additional options. For functional enrichment, genes with an FDR $\leq$ 0.1, or, if only a few genes were below this FDR, top genes with the smallest p values, as indicated in the results section, were used.

In our general bacterial experiments, data were pre-filtered by removing genes that did not have a cpm >3 in at least two samples (replicates). Differential gene expression was calculated applying Genewise Negative Binomial Generalized Linear Models with Quasi-likelihood Tests (edgeR packages glmQLFit and glmQLFTest). For our infection experiments, we used the more rigorous function glmTreat (edgeR package) to find differentially expressed genes (FDR $\leq$ 0.05) with a fold change significantly higher than 1.5 fold in the motility mutants that might affect phagocytic uptake. Dispersions and generalized linear model fits (R functions estimateDisp and glmQLFit) were calculated using the robust option.

## Data analyses – interspecies correlation

The differential gene expression of 12298 expressed host genes (one cpm in at least one sample) was calculated for each individual infected replicate compared to the non-infected host-cell group, resulting in $\log_2$FC data of eight samples (PA14, Δ*fliC*, Δ*flgK*, and Δ*motABCD*, each in two replicates). On the bacterial site, we used the 53 genes that were differentially expressed in all three flagella mutants as compared to PA14 upon host-cell contact (see *Figure 4*). To determine which genes of the infected host cells correlate with bacterial in vivo gene expression, bacterial reads per gene were normalized using cpm and the weighted trimmed mean of M-values with singleton pairing (TMMwsp, edgeR functions calcNormFactors and cpm). To determine the genes of the infected host cells, whose expression correlate with the bacterial response to the host, the differential gene expression (see above) of all in vivo versus planktonic bacteria was calculated to obtain $\log_2$FC data

for each individual replicate (as for the host). For each of the 53 bacterial genes, host genes with a Spearman correlation coefficient $\geq |0.8|$ were counted for all expressed host genes, as well as for the 74 expressed $PIP_3$-associated genes (from the KEGG pathway mmu04070). Hypergeometric tests (R-function phyper) were used to calculate whether the proportion of $PIP_3$-associated genes (that correlated with the individual bacterial genes) was significantly overrepresented. Multiple testing was corrected using FDR.

### Data analyses – visualizations
Heatmaps were created with the R package superheat (*Barter and Yu, 2018*). Barplots with functional enrichment were visualized with ggplot2 (*Ginestet, 2011*). Venn diagrams were done using the euler function of the R package eulerr (*Larsson, 2018*). We also analyzed whether the number of genes in the Venn diagrams are significantly different than expected by chance. For this purpose, we permuted data sets with the same numbers of genes that were used for the Venn diagram. We considered the groups as significant if the value was higher in maximum 5% of the permutations.

### Data analyses – statistics
Significance between two groups was determined using the nonparametric Mann-Whitney test, while one-way ANOVA with Bonferroni posttest was used to compare two or more groups. Significance levels of $p<0.05$, $p<0.01$, or $p<0.001$ were denoted with asterisks: *, **, and ***, respectively.

### Data resources
Raw files and reads per gene of the RNA sequencing approaches have been deposited in the NCBI Gene Expression Omnibus (GEO) under the accession number GSE141757.

## Acknowledgements
We gratefully thank Anja Kobold and Astrid Dröge for their expert technical assistance and the Genome Analytics Group (GMAK) from the HZI, Braunschweig for their assistance with the dual-sequencing samples. SH was funded by the EU (ERC Consolidator Grant COMBAT 724290) and received funding from the Deutsche Forschungsgemeinschaft (DFG, German Research Foundation) under Germany's Excellence Strategy – EXC 2155 'RESIST' – Project ID 390874280 and in the frame of the SPP 1879.

## Additional information

### Funding

| Funder | Grant reference number | Author |
| --- | --- | --- |
| H2020 European Research Council | COMBAT Project 724290 | Susanne Häussler |
| Deutsche Forschungsgemeinschaft | Germany's Excellence Strategy - EXC 2155 RESIST. 390874280 | Susanne Häussler |
| DFG | SPP 1879 | Susanne Häussler |

The funders had no role in study design, data collection and interpretation, or the decision to submit the work for publication.

### Author contributions
Sebastian Felgner, Matthias Preusse, Conceptualization, Resources, Supervision, Validation, Investigation, Data curation, Software, Visualization, Methodology, Writing - original draft, Project administration, Writing - review and editing; Ulrike Beutling, Conceptualization, Data curation, Software, Validation, Investigation, Visualization, Methodology, Writing - original draft, Writing - review and editing; Stephanie Stahnke, Software, Validation, Investigation, Visualization, Methodology, Writing - review and editing; Vinay Pawar, Conceptualization, Validation, Investigation, Visualization, Methodology, Writing - review and editing; Manfred Rohde, Mark Brönstrup, Conceptualization, Resources,

Validation, Investigation, Visualization, Methodology, Writing - review and editing; Theresia Stradal, Conceptualization, Resources, Supervision, Validation, Methodology, Project administration, Writing - review and editing; Susanne Häussler, Conceptualization, Resources, Supervision, Funding acquisition, Methodology, Writing - original draft, Project administration, Writing - review and editing

### Author ORCIDs
Sebastian Felgner ⬤ https://orcid.org/0000-0003-0030-2490
Matthias Preusse ⬤ https://orcid.org/0000-0002-2775-3139
Mark Brönstrup ⬤ http://orcid.org/0000-0002-8971-7045
Susanne Häussler ⬤ https://orcid.org/0000-0001-6141-9102

### Ethics
Animal experimentation: All animal experiments were performed according to guidelines of the German Law for Animal Protection and with permission of the local ethics committee and the local authority LAVES (Niedersächsisches Landesamt für Verbraucherschutz und Lebensmittelsicherheit). C57BL/6 mice derived from our own breeding.

### Decision letter and Author response
Decision letter https://doi.org/10.7554/eLife.55744.sa1
Author response https://doi.org/10.7554/eLife.55744.sa2

## Additional files

### Supplementary files
• Transparent reporting form

### Data availability
Raw files and reads per gene of the RNA sequencing approaches have been deposited in the NCBI Gene Expression Omnibus (GEO) under the accession number GSE141757.

The following dataset was generated:

| Author(s) | Year | Dataset title | Dataset URL | Database and Identifier |
|---|---|---|---|---|
| Felgner S, Preusse M, Beutling U, Stahnke S, Pawar V, Rohde M, Brönstrup M, Stradal T, Haeussler S | 2020 | Host-induced spermidine production in motile *Pseudomonas aeruginosa* triggers phagocytic uptake | https://www.ncbi.nlm.nih.gov/geo/query/acc.cgi?acc=GSE141757 | NCBI Gene Expression Omnibus, GSE141757 |

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
