## [Decision Letter]

**Acceptance summary:**

In this elegant study, the authors report the reasons for the known lower immune response and phagocytic uptake of non-motile mutant of *Pseudomonas aeruginosa* by macrophages. Using an ingenuous combination of mutants and dual RNA-seq approach the authors dissected that motile bacteria produce spermidine when encountering host cells, and such spermidine drives the upregulation of the PIP_3_ host response. The overall message of the paper reveals new insights into the biology of *Pseudomonas* and Gram-negative infections.

**Decision letter after peer review:**

Thank you for submitting your article "Host-induced spermidine production in motile *Pseudomonas aeruginosa* triggers phagocytic uptake" for consideration by *eLife*. Your article has been reviewed by three peer reviewers, and the evaluation has been overseen by Gisela Storz as the Senior Editor and a Reviewing Editor. The reviewers have opted to remain anonymous.

The reviewers have discussed the reviews with one another and the Reviewing Editor has drafted this decision to help you prepare a revised submission. Please aim to submit the revised version within two months or as soon as the current situation will allow you to carry out wet experimental work.

In this elegant and well written study, you report your investigations of the reasons for the known lower immune response and phagocytic uptake of non-motile mutant of *Pseudomonas aeruginosa* by macrophages. Using an ingenuous combination of mutants and dual RNA-seq approach you dissected that motile bacteria produce spermidine when encountering host cells. You also show that spermidine drives the upregulation of the PIP_3_ host response. The overall message of the paper is appealing, simple and clear and unpacks new insights into the biology of *Pseudomonas* and Gram-negative infections.

The reviewers agreed that the following would help to further improve the study:

Essential points:

1) In addition to bacterial in vitro cultures in rich medium (LB OD 2) as a reference control for the dual RNA-seq experiment, reviewers would like to see an additional bacterial reference control, e.g. RNA from bacteria grown in LB to OD 2, but then shifted to DMEM and incubated therein for 3 h at 37°C (in the absence of host cells). Comparison to such a control would then really allow to infer the host-specific expression profile of the different *Pseudomonas* strains.

2) PA14 is generally considered to be cytotoxic with the T3SS playing a role in killing. Are the cells all surviving at the time point of the dual RNA-seq experiment? Please perform a lactate dehydrogenase or Pi uptake assay to compare the effect of the panel of mutants on cytotoxicity.

3) Please try to improve the dual RNA-seq analysis by carrying out interspecies correlation analysis of the host and bacterial responses to the different infections (different mutants). This will allow to check that there is nothing else than spermidine influencing the host response and conversely other pathways than PIP_3_. In addition, it will be a more compelling analysis than stating "the top 100" or the "top 1000" genes and then focussing on a few.

[Editors' note: further revisions were suggested prior to acceptance, as described below.]

Thank you for re-submitting your revised article "Host-induced spermidine production in motile *Pseudomonas aeruginosa* triggers phagocytic uptake" for consideration by *eLife*. Your article has been reviewed by the same three peer reviewers and the evaluation has been overseen by Gisela Storz as the Senior Editor and a Reviewing Editor. The reviewers have opted to remain anonymous.

The reviewers unanimously agree that you have significantly improved the manuscript and would like to thank you for your efforts. However, they also all agree that the following is essential before they can consider acceptance of the manuscript:

– Additional bacterial reference control, e.g. RNA from bacteria grown in LB to OD 2, but then shifted to DMEM and incubated therein for 3 h at 37C (in the absence of host cells) is needed. In absence of such a control, the host-specific expression profile of the different *Pseudomonas* strains cannot be ascertained and the paper would lose a lot of its novelty. Reviewers were sympathetic that such experiment takes more time because of the current crisis, but had all agreed, in agreement with the journal to lift the time constraint.

---

## [Author Response]

Essential points:1) In addition to bacterial in vitro cultures in rich medium (LB OD 2) as a reference control for the dual RNA-seq experiment, reviewers would like to see an additional bacterial reference control, e.g. RNA from bacteria grown in LB to OD 2, but then shifted to DMEM and incubated therein for 3 h at 37°C (in the absence of host cells). Comparison to such a control would then really allow to infer the host-specific expression profile of the different Pseudomonas strains.

The reviewers raised an interesting point. In our study, we concentrated on the elucidation of the molecular mechanisms underlying the differences in phagocytic uptake of motile and non-motile bacteria. We recorded the transcriptional profiles of the wild-type as well as non-motile mutants and observed substantial transcriptional differences between the groups. To demonstrate that this transcriptional reprogramming is not due to the non-/ motility per se, we recorded the transcriptional profiles of the wild-type and the motility mutants also in LB as a standard medium. Here, no major differences in the transcriptional profiles were recorded. We think this is a very important control. We elaborated in the revised version of the manuscript in more detail, how we use the LB transcriptional profiles as a base-line to describe not only a differential gene expression in the motility mutants as compared to the PA14 wild-type strain upon host cell contact, but to also provide information on whether the differentially expressed gene was induced in the wild-type or the mutant upon host cell contact in comparison to the basal LB gene expression profile.

Nevertheless, we agree with the reviewers that, if the transcription profiles would have been recorded in cell culture medium, this would have allowed to infer that the observed differential in spermidine production between the strains is not the result of their differential response to cell culture medium. Although we cannot rule out, that the bacteria respond to something specific in the cell culture medium (HEPES, glucose as sole carbon source etc.), spermidine production in PA14 is also not induced in BM2 medium. Furthermore, FCS is not inducing spermidine production, as spermidine production in PA14 was shown to be induced also in FSC-free cell culture medium in the presence of host cells.

To make this clear we added the following sentence to the revised version of the manuscript: “Of note, the motility variants differentially expressed genes encoding these structures exclusively under the infection experiment conditions – most likely due to host-cell contact – and not under standard lab (LB or BM2 growth) conditions.”

2) PA14 is generally considered to be cytotoxic with the T3SS playing a role in killing. Are the cells all surviving at the time point of the dual RNA-seq experiment? Please perform a lactate dehydrogenase or Pi uptake assay to compare the effect of the panel of mutants on cytotoxicity.

We completely agree with the reviewers. It is extremely important to guarantee comparable infection conditions between the mutants and the wild-type strain. Thus, we performed an LDH assay as requested by the reviewer. No differences were observed between PA14 Wt, *ΔfliC* and *ΔflgK* 3 h post infection. The increased values of the *ΔmotABCD* mutant may derive from the low contact between the host and the bacteria as demonstrated with the adherence assay. Thus, the bacteria could proliferate better leading to a higher effective bacterial dose over time in comparison to the other strains. The results were added to the new Figure 3—figure supplement 2.

The LDH assay was added to the Material and methods section (subsection “Invasion assays”) and the results were included in the revised version of the manuscript (see subsection “Dual-sequencing approach to analyze host-pathogen interactions”).

3) Please try to improve the dual RNA-seq analysis by carrying out interspecies correlation analysis of the host and bacterial responses to the different infections (different mutants). This will allow to check that there is nothing else than spermidine influencing the host response and conversely other pathways than PIP_3_. In addition, it will be a more compelling analysis than stating "the top 100" or the "top 1000" genes and then focussing on a few.

We performed the interspecies correlation analysis as requested by the reviewers. We were able to confirm the role of *arnT* and the spermidine synthesis genes, as well as of *pmrA* and *phoPQ* during infection. Expression of these genes positively correlated (rho ≥ 0.8) with the expression of significantly more PIP_3_ associated host genes as we would expect by chance (hypergeometric test, see subsection “Data analyses – Interspecies correlation”). Thus, the correlation analysis underscores the experimental findings and we added the results of this analysis in the revised version of the manuscript. The data are presented in Figure 4—source data 2 and Figure 4—source data 3.

[Editors' note: further revisions were suggested prior to acceptance, as described below.]

However, they also all agree that the following is essential before they can consider acceptance of the manuscript:– Additional bacterial reference control, e.g. RNA from bacteria grown in LB to OD 2, but then shifted to DMEM and incubated therein for 3 h at 37°C (in the absence of host cells) is needed. In absence of such a control, the host-specific expression profile of the different Pseudomonas strains cannot be ascertained and the paper would lose a lot of its novelty. Reviewers were sympathetic that such experiment takes more time because of the current crisis, but had all agreed, in agreement with the journal to lift the time constraint.

We performed the transcriptional profile for the medium switch experiment as suggested by the reviewers. Briefly, bacterial cultures were grown in LB for 3.5 h and adjusted to the infection dose of 5*10^6 bacteria/ml in order to keep the experimental conditions similar to that of the infection experiments with host cells. The adjusted bacteria were washed, transferred into DMEM and grown for 3 h at 37°C. Afterwards, RNA extraction, downstream processing and data analysis has been done as for the dual-sequencing experiment. The experimental procedure of the medium switch experiment has been added to the Materials and methods section.

The data demonstrates that neither the spermidine-associated genes nor the *arnT* operon is significantly regulated by switching the medium when comparing PA14 to its *ΔfliC* variant. The results are included in the revised version of the manuscript in subsection “Non-motile *P. aeruginosa* have altered expression levels of *arnT* and of genes involved in spermidine and pyoverdine biosynthesis” and in a new table Figure 4—source data 4. Of note, the results are also showing that obviously the differential expression of the pyoverdine biosynthesis operon between PA14 to its *ΔfliC* variant is becoming already apparent upon switching the cultures to cell culture medium and thus is not exclusively the result of host cell contact. We added a statement regarding the differential expression of pyoverdine genes upon medium switch in subsection “Non-motile *P. aeruginosa* have altered expression levels of *arnT* and of genes involved in spermidine and pyoverdine biosynthesis” and the Discussion.